# The Protein Composition Changed the Quality Characteristics of Plant-Based Meat Analogues Produced by a Single-Screw Extruder: Four Main Soybean Varieties in China as Representatives

**DOI:** 10.3390/foods11081112

**Published:** 2022-04-13

**Authors:** Bo Lyu, Jiaxin Li, Xiangze Meng, Hongling Fu, Wei Wang, Lei Ji, Yi Wang, Zengwang Guo, Hansong Yu

**Affiliations:** 1College of Food Science and Engineering, Jilin Agricultural University, Changchun 130118, China; michael_lvbo@163.com (B.L.); li1997jiaxin@163.com (J.L.); mxz625797@163.com (X.M.); 15764381475@163.com (H.F.); jilei0616@163.com (L.J.); wangyi284419@163.com (Y.W.); 2Division of Soybean Processing, Soybean Research & Development Center, Chinese Agricultural Research System, Changchun 130118, China; wangwei19936@163.com; 3College of Food Science, Northeast Agricultural University, Harbin 150030, China; 4Jilin Provincial Agricultural Products Processing Industry Promotion Center, Changchun 130022, China

**Keywords:** extrusion technology, textured soy protein, protein subunit composition, processing applicability, plant-based meat analogues

## Abstract

Plant-based meat analogues (PBMs) are increasingly interesting to customers because of their meat-like quality and contribution to a healthy diet. The single-screw extruder is an important method for processing PBMs, and the characteristics of the product are directly affected by the composition of the raw materials; however, little research focuses on this issue. To explore the effect of protein composition on the quality characteristics of PBMs produced by a single-screw extruder, four soybean varieties used in China (Heihe 43 (HH 43), Jiyu 86 (JY 86), Suinong 52 (SN 52), and Shengfeng 5 (SF 5)) were selected. The 11S/7S ratios for these varieties ranged from 1.0: 1 to 2.5: 1 in order to produce PBMs with different protein compositions. The structure, processing, nutrition, and flavor characteristics were explored to analyze their differences. The results showed that protein composition affected the structure of PBMs, but the correlation was not significant. Meanwhile, a lower 11S/7S ratio (HH 43) did not prove to be a favorable characteristic for the processing of PBMs. From the perspective of nutrition and flavor, it seems acceptable to use a moderate 11S/7S ratio (JY 86 and SN 43) to produce PBMs. This study proved that the protein composition of raw materials affects the characteristics of PBM products produced by a single-screw extruder. To produce PBMs of higher quality, soybeans with a markedly different 11S/7S ratio should not be selected.

## 1. Introduction

With improvements in living standards, great changes have taken place in people’s dietary structures, including the increased intake of animal-derived foods [1]. However, the increase in the intake of animal-derived foods and the decrease in vegetarian intake is one of the causes of many diseases, such as intestinal and cardiovascular diseases [2,3]. This result may be caused by the excessive intake of fats, drug residues, or other factors [4,5]. To maintain the excellent taste and good processing characteristics of meat products while preventing the potential health risks caused by the excessive intake of animal-based food, plant-based meat analogues (PBMs) came into being. The intake of PBM products not only does not cause health risks but also can reduce land use and resource consumption [6,7]. PBMs can also meet the requirements of modern people for food diversity [8]; this has gradually increased the acceptability of PBM products, as well as the demand for them [9].

An important aspect of PBM products is the fact that the process of manufacturing textured plant protein mainly depends on the changes in protein structure caused by the high temperature and pressure produced by screw extruders [10]. Mainstream textured plant protein production equipment includes the twin-screw extruder and the single-screw extruder [11]. The twin-screw extruder is widely used because of its excellent processing capacity; as a result, there are more theoretical studies based on it [10,12]. However, the single-screw extruder is also used on a large scale because of its lower cost and ability to process a wider variety of raw materials, such as insoluble dietary fiber, starch, etc. [13]. Therefore, consumers may encounter products manufactured using two different kinds of extrusion technologies in the market at the same time. Certainly, there are differences in the quality of the products because of the differences in chemical cross-linking and molecular aggregation [14].

Soybean protein is considered to be a good choice for producing PBMs because of its excellent gelation, superior nutritional value, low cost, and safety as a raw material [15]. The composition of soybean protein affects the processing and nutritional characteristics of soybean products directly [16]. In particular, the proportions of soy glycinin (11S) and soy β-conglycinin (7S) directly affect the key processing characteristics, such as gel and foaming properties [17,18]. It is generally believed that high 7S content is related to hydration characteristics, such as emulsification and foaming properties [17,19]. A high 11S content means a higher protein structural strength, as manifested in characteristics such as gelation [20]. However, the relationship between them is not strictly linear. There is reason to believe that the texture of PBMs is also related to the gel properties of soybean protein [21]. However, no study has compared the properties of PBM products produced using different varieties of soybean with different protein compositions, and no study has demonstrated what kind of protein composition is more suitable for PBM products produced by single-screw extruders. Previous studies on PBMs tend to analyze the composition of soybean protein isolate (SPI). Currently, under the guidance of “Whole-Soybean Processing”, a great deal of PBMs are processed directly from defatted soybean powder. As such, we should pay more attention to the protein composition of soybean. In addition, soybean protein should be perceived as a safe raw material for producing PBM products. Soybeans used in the production of protein products are required not to use transgenic soybeans in many countries [22,23,24]; similar regulations also avoid some of the risks of processing livestock products, such as hormones [25]. Although soybean protein has a certain potential to aggravate food sensitivities, high temperatures and pressures, such as those the single-screw extruder provides, can reduce the allergen content [26], which can improve the acceptability of soybean products. Therefore, soybean protein can be regarded as the best raw material for the production of PBMs.

In this study, to explore the potential impact of the soybean protein composition of the raw materials on PBM characteristics, we selected four soybean varieties, representative of the main planting varieties in China, with different compositions and ratios of 11S and 7S (11S:7S = 1, 1.5, 2, 2.5), which were processed into defatted soybean flours (DSFs) with different protein compositions. A single-screw extruder was used to produce textured soybean protein as a representative of PBM products. The structure (basic composition, sulfhydryl content, secondary structure, functional group composition, and microstructure), processing characteristics (water-absorption capacity, water-holding capacity, water-swelling capacity, tensile strength, breaking elongation, and texture characteristics), nutritional characteristics (dietary fiber, reducing sugar, phytic acid, trypsin inhibitor, plant lectin, amino acid composition, and isoflavone content) and flavor characteristics were used to measure the quality of PBM products, to determine the impact of different protein compositions on their quality.

## 2. Materials and Methods

### 2.1. Selection of Soybean Varieties and Preparation of Defatted Soybean Flour (DSF)

By searching the database of the China Agriculture Research System (CARS): Soybean Processing Division, we selected four kinds of soybeans with different protein subunit compositions as the experiment raw materials: Heihe 43 (HH43), Jiyu 86 (JY 86), Suinong 52 (SN 52), and Shengfeng 5 (SF 5), for which the ratios of 11S/7S were 1.0, 1.5, 2.0, and 2.5, respectively. All four varieties are under large-scale cultivation in China. After soybean dehulling, the oil was removed by an oil press, and then defatted soybean flour (DSF) was obtained after crushing (oil content <7%). A SH-28 single-screw extruder (Shandong Yuya Soybean Machinery Manufacturing CO., Ltd., Zaozhuang, China) was used to produce PBMs under the following conditions: the ratio of DSF to water was 2:3; the temperature was 240 °C in the first zone, 220 °C in the second zone, 200 °C in the third zone, and 180 °C in the fourth zone; the screw rotated at 70 rpm.

### 2.2. Analysis of Soybean Protein Composition by SDS-PAGE

The analysis of the soybean protein composition was measured according to the method described by Song et al. [27]. The presence and absence of glycinin and β-conglycinin subunits in the soybean seeds were confirmed by SDS-PAGE. The total seed proteins were extracted from a small portion of cotyledon tissues with an SDS sample buffer (2% SDS, 5% 2-mercaptoethanol, 10% glycerol, 5 M urea, and 62.5 mM Tris aminomethane) and then centrifuged at 15,000× *g*. Then, 10 µL of the supernatant was separated on 4.5% stacking and 12.5% separating polyacrylamide gels and stained with Coomassie Brilliant Blue R250. The gels were scanned by an Image Lab 3.0 (Bio-Rad Laboratories, Inc., Hercules, CA, USA), and the number of protein subunits was quantified according to the gray value.

### 2.3. The Structure of PBMs

#### 2.3.1. The Basic Composition of Four Kinds of PBMs

The moisture, protein, ash, and fat content of four kinds of PBMs were determined according to the AOAC Official Method (AOAC 2007.04).

#### 2.3.2. The Sulfhydryl Content of Four Kinds of PBMs

Four kinds of PBMs were crushed into 80 mesh after being fully dried, and a method employing Ellman’s reagent (10 mm DTNB, 0.2 mm EDTA) was used to determine the content of SH in the samples [28]. The preparation of Tris-Glycine buffer was as follows: 0.086 M tris + 0.09 M Glycine + 4 mM EDTA, pH = 8.0. The samples were dispersed in the Tris-Glycine buffer to obtain 2 mg/mL solutions. Then, 0.03 mL Ellman’s reagent was added to 3 mL solution, and the solution was mixed immediately and stored for 15 min at room temperature before measuring the absorbance at 412 nm. A buffer solution without a protein sample was used as a reagent blank.

#### 2.3.3. Circular Dichroism Spectrum (CD)

The crushed PBM samples (80 mesh) were prepared with 1 mM phosphate-buffered saline (PBS, pH = 7.0) into a 1 mg/mL solution, placed in a 1 mm optical path length quartz cell, and measured with a J-810 CD spectrometer (JASCO, Tokyo, Japan). The sensitivity was set to 2 mdeg/cm. The 185–260 nm CD spectrum was recorded at 20 °C.

#### 2.3.4. Fourier Transform Infrared Spectroscopy (FT-IR)

Fourier transform infrared spectroscopy (FT-IR) analyses of four kinds of PBMs were performed using a Nicolet iS5 spectrometer (Thermo Fisher, Waltham, MA, USA). The dried samples were mixed with KBr powder (1:100, *w*/*w*) after being crushed into 80 mesh, and the spectra were read over the range of 4000–400 cm^−1^ with a resolution of 4 cm^−1^.

#### 2.3.5. Scanning Electron Microscopy (SEM)

The PBMs were cut into strips of appropriate size for sample pre-treatment. The samples were immersed in a glutaraldehyde solution (2.5%, pH = 7.2–7.4) for 24 h. After they were fixed, the samples were washed with a phosphate buffer (pH = 7.2) 3 times before eluting with 30%, 50%, 70%, 85%, 95%, and 100% ethanol solutions. The samples were made into 2 mm × 2 mm flakes, as thin as possible, after freeze-drying in order to observe the microscopic appearance of four kinds of PBMs using a SU8020 scanning electron microscope (SEM; Hitachi, Tokyo, Japan) after spraying with a gold–palladium alloy. The scanning images were captured at accelerating voltages of 5 kV and photographed at magnifications of 5000X (scale bar 10 μm).

### 2.4. The Processing Characteristics of PBMs

#### 2.4.1. Water-Absorption Capacity (WAC), Water-Holding Capacity (WHC), and Water-Swelling Capacity (WSC)

WAC: After recording the weight of the fully dried sample (M1), it was soaked in 60 °C water for 5 h, then drained for 6 min. The samples were weighed (M2).
WAC = (M2 − M1)/M1 × 100% (1)

WHC: 1.00 g crushed sample (M1) and 20 mL of water were mixed in a dry centrifuge tube (M0). The sample was kept at RT for 24 h and centrifuged at 4000 rpm for 20 min. The supernatant was removed, and the weight (M2) was recorded.
WHC = (M2 − M0)/M1 × 100% (2)

WSC: 1.000 g crushed, fully dried sample (M) and 10 mL of water were mixed in a dry centrifuge tube and kept at RT for 24 h. The volume of the sample was recorded (V).
WSC = V/M × 100% (3)

#### 2.4.2. The Tensile Strength (TS) and Breaking Elongation (BE) of Four Kinds of PBMs

The WDW-200H electronic tensile testing machine (Hongtuo, Dongguan, China) was used to analyze the tensile strength of four kinds of PBMs. The experiment conditions were as follows. The wet PBM was cut to 10 cm × 6 cm for testing; the initial clamping distance was 40 mm, and the tensile speed was 5 mm/s. The following values were recorded: the maximum tension at break (P), the cross-sectional area of samples (S), the elongation at break (δL), and the original length (L). Values were calculated for the tensile strength (TS) and breaking elongation (BE).
TS (MPa) = P/S (4)
BE (%) = δL/L × 100% (5)

#### 2.4.3. Texture Profile Analysis (TPA)

The texture characteristics of four kinds of PBMs were analyzed by a Texture Analyzer (TA.new plus, Isenso, Shanghai, China). The samples were cut into a square with a length of 10 mm, and the conditions were as follows. The detection mode was TPA mode with the P/36R detector; the rate before the test was 2 mm/s; the rate during the test was 1 mm/s; the rate after the test was 2 mm/s; the compression degree was 50%.

### 2.5. The Nutritional Characteristics of PBMs

#### 2.5.1. The Dietary Fiber, Reducing Sugar, Phytic Acid, Trypsin Inhibitor, Plant Lectin, and Isoflavone Content of Four Kinds of PBMs

The dietary fiber, reducing sugar, phytic acid, and isoflavone content of four kinds of PBMs were determined according to the AOAC Official Methods (AOAC 2017.16, AOAC 945.66, and AOAC 986.11). The amounts of trypsin inhibitor and plant lectin (Soybean agglutinin, SBA) found in the four kinds of PBMs were determined using the Trasylol Elisa kit (Ml064289, Enzyme-linked Biotechnology Co., Ltd., Shanghai, China) and the SBA Elisa kit (Ml002453, Enzyme-linked Biotechnology Co., Ltd., Shanghai, China).

#### 2.5.2. The Amino Acid Composition

The analysis of the amino acid composition of four kinds of PBMs was performed according to the method described by Song et al. [27]. After the PBMs were fully dried, a meal was prepared by mill grinding through a 0.25-mm sieve and thoroughly mixing. Total amino acids were obtained by the hydrolysis of seed meal with an excess of 6 M HCl for 22 h in sealed evacuated tubes at a constant boiling temperature (110 °C). An L-8800 amino acid analyzer (Hitachi, Tokyo, Japan) was used to determine the amino acid compositions of the hydrolysates. The amino acid composition was expressed as relative content (%) on a dry basis.

### 2.6. The Flavor Characteristics of PBMs

The volatile flavor compounds found in four kinds of PBMs were measured by a 6890N-5975B Gas Chromatography-Mass Spectrometry workstation (GC-MS; Agilent, Palo Alto, CA, USA).

Sample pre-treatment was as follows. The samples were sealed in head-space bottles and warmed at 80 °C in a water bath for 30 min. A solid-phase microextraction needle (SPMEN; 100 μL PDMS; SUPELCO, Bellefonte, PA, USA) was used for extraction for 30 min at 80 °C before desorption for 5 min.

The experiment conditions for GC-MS were as follows. Chromatographic column: HP-5MS (30 m × 0.25 mm × 0.25 μm); split ratio: no split; carrier gas flow rate: 1.2 mL/min; injection port temperature: 250 °C; scanning mode: full scan; ion source temperature: 230 °C; quadrupole temperature: 150 °C; temperature program: initial temperature of 50 °C for 2 min, raised to 180 °C at the rate of 5 °C/min for 5 min, then raised to 250 °C at the rate of 10 °C/min for 5 min.

The mass spectra were searched using the NIST database to identify the volatile components in the samples, and the relative content of each component was analyzed by the area normalization method.

### 2.7. Statistical Analysis

All determinations were conducted at least three times, and all results were expressed as mean ± standard deviation ($\overset{¯}{x}$ ± SD). One-way analysis of variance (ANOVA) and Duncan’s test were used to analyze the differences in the properties of four kinds of PBMs using IBM SPSS 25.0 (SPSS Inc., Chicago, IL, USA); *p* < 0.05 was considered significant, and all results were expressed as mean ± standard deviation ($\overset{¯}{x}$ ± SD). All statistical graphs were produced with Origin Pro 2018 (GraphPad Software Inc., San Diego, CA, USA).

## 3. Results and Discussion

### 3.1. Effect of Protein Composition on the Structural Characteristics of PBMs

The basic compositions of the four kinds of PBMs are shown in Table 1. Among them, the protein (49.18%) and oil contents (4.17%) of SF 5 were slightly higher than those of the other varieties. The SDS-PAGE spectra of four kinds of PBMs are shown in Figure 1a, and the quantitative results of different protein subunits are shown in Table 2. The content and ratio of 11S/7S differed significantly among the four kinds of PBMs, of which SF 5 showed the highest ratio of 11S/7S (2.50), and HH 43 showed the lowest (1.05). This result is consistent with the protein composition of the corresponding soybeans in the CARS database, which met the requirements of the experiment.

The FTIR spectra of four kinds of PBMs are shown in Figure 1b. As shown, the significant absorption peaks were located at 3281 cm^−1^, 2928 cm^−1^, 2850 cm^−1^, 2366 cm^−1^, 1740 cm^−1^, 1632 cm^−1^, 1527 cm^−1^, 1247 cm^−1,^ and 1042 cm^−1^; the spectra of the four kinds of PBMs showed minor differences. The spectra are determined by the combination of protein and dietary fiber in PBMs. The secondary structure of the protein was determined based on the amide I band analysis (1700–1600 cm^−1^) [29], but there are also some functional groups similar to soybean dietary fiber in other components, such as some aldehyde and carboxyl groups [30]. Thus, the functional group composition of the four PBMs displayed little difference.

The effect of the protein composition on free sulfhydryl (SH) content in PBMs is shown in Figure 1c. The SH content seems to have no obvious correlation with protein composition: HH 43 showed the highest, and JY 86 was the lowest. The SH content of 11S is higher than that of 7S [31]. However, in a mixed system, more 11S converts SH to disulfide bonds [32]. Therefore, this result might be caused by the mixed nature of the system and the protein denaturation process of PBMs. Studies have shown that the SH content in textured protein is related not only to raw materials but also to extrusion temperature, protein denaturation, and other factors [33,34]. Studies have shown that, with the extrusion process, the degree of protein cross-linking increases, the proportion of high-molecular-weight protein subunits increases, and the small-molecular-weight subunits decreases, resulting in a lower SH content [35]. Therefore, the high SH content does not mean that the texture characteristics of the PBMs were better; this needs to be discussed comprehensively in combination with future research.

The CD spectra of four kinds of PBMs are shown in Figure 1d. Combined with the analysis of the amide I band (1700–1600 cm^−1^) in FTIR, the secondary structure compositions of the proteins in the four kinds of PBMs are shown in Table 3. There were significant differences in the protein secondary structures of the four kinds of PBMs. Compared to conventional soybeans or soy protein isolate (SPI), the ratios of the various secondary structures were also different [29,36]. The secondary structures of soybean proteins of different varieties should be very different, but the differences between them are significantly smaller after being processed into PBM. This shows that the screw extrusion process rearranges the secondary structures of the proteins, which should be regarded as the key factor for changing the secondary structures of proteins, rather than choosing different soybean varieties. In proteins, the existence of α-helix and β-sheet is mainly maintained by hydrogen bonds, while β-turn depends on the amino acid residues with charge [37]. This result shows that the extrusion process strengthens the rigidity of the protein structures, reduces the exposure to amino acid residues, and makes the protein structures more stable. A similar result has been found in other studies [38,39].

The outward appearances of the four kinds of PBMs are shown in Figure 1f. As shown, the appearances of the four kinds of PBMs were not very different; among them, the surface of HH 43 was slightly dense. It should be noted that the wrinkles on the surface of PBMs result from the shear caused by the grinding head changing between different stages, rather than any difference in the apparent structure of the PBM itself. Figure 1e shows the differences in the microstructures of the four kinds of PBMs. JY 86 has an obvious lamellar structure and flat surface (II), the surface of HH 43 is uneven with torn lamellae (I), SN 52 has a smooth surface, less clearance, and an irregular shape (III), and SF 5 has the most obvious structure and is dense with an irregular shape (IV).

In summary, protein composition did affect the structural properties of the PBMs, but the correlation was not significant.

### 3.2. Effect of Protein Composition on the Processing Characteristics of PBMs

The water absorption capacity (WAC, WHC, and WSC) of the four kinds of PBMs is shown in Figure 2a. As shown, JY 86 showed the highest WAC (275%); a lower or higher ratio of 11S/7S results in a lower WAC. Meanwhile, WHC and WSC had a positive correlation with the 11S/7S ratio, wherein a higher content of 11S led to a higher WHC and WSC. Excellent water adsorption capacity could make PBMs more like meat [40]. Therefore, the higher WHC (274%) and WSC (135%) might give SF 5 superior processing characteristics.

Tensile strength (TS) refers to the maximum tensile capacity that food can bear and represents the toughness of food. Breaking elongation (BE) is the maximum length change of food before fracture and represents the elasticity. The results are shown in Figure 2b,c. The trends for TS and BE were similar: SN 52 (11S/7S = 2.0) showed the highest forward strength (TS = 0.013 MPa, BE = 23.38%) and JY 86 (11S/7S = 1.5) showed the highest reverse strength (TS = 0.030 MPa, BE = 32.41%). The forward strength represents the tightness of the textured structure of the PBM, while the reverse strength represents the strength of the force between proteins. Therefore, JY 86 might have better toughness and elasticity. In addition, this result seems to be opposite to that of free SH content in Figure 1c, in that the PBM with the lowest SH content showed the best elasticity and toughness; this proves that a high SH content does not mean that the texture characteristics of a PBM are better.

The texture characteristics of four kinds of PBMs were determined by a texture analyzer, and the results are shown in Table 4. The results show that the five indexes were directly proportional to the 11S content; that is, SF 5 showed the best texture characteristics. Among them, the resilience and the springiness increased gradually as 11S content increased, but there was no significant difference (*p* > 0.05), whereas the hardness, adhesiveness, and chewiness were significantly different among different varieties (*p* < 0.05). This showed that 11S globulin plays an important role in the extrusion process.

The 11S content is closely related to the textural properties of soy products. For protein gel especially, there is a correlation between 11S and textural properties [41]. Increasing the 11S content could improve the texture quality of protein products to a certain extent, which is attributable to the formation of disulfide bonds [42]. A study by Zheng et al. showed that a higher β sheet content and a high ratio of 11S/7S increases the quality of soy protein gel, and disulfide bonds might be one of the reasons [43]. In addition, in the process of soy texturization, the processing conditions also strengthen the texture characteristics of the product, such as temperature and pressure, among others. Research has shown that, in conventional soy product processing, adjusting the pressure and temperature of raw material processing directly affects the processing characteristics of the products [44]; this is caused by changes in the solubility, conformation, and protein aggregation of 11S. in the same is true of the extrusion process. During heating, individual subunits within globulins undergo dissociation, unfolding, and reaggregation to render them more functional by virtue of qualities such as higher gelation [45]. In essence, textured protein is another form of gelation, which also requires the rearrangement of different protein subunits. The higher 11S content also leads to the higher strength of the protein aggregates [46], which better withstand high pressure [47]. Therefore, the excellent texture characteristics of SF 5 may be due to the high 11S content; meanwhile, the high temperature and high pressure provided by the extrusion process enhance the rearrangement of the protein subunits and the formation of the spatial structure.

### 3.3. Effect of Protein Composition on the Nutritional Properties of PBMs

We chose to use the content of dietary fiber, reducing sugar, phytic acid, phytohemagglutinin, trypsin inhibitor, isoflavone, and amino acid composition to analyze the nutritional characteristics of PBMs comprehensively. The results are shown in Figure 3 and Figure 4 and Table 5.

The dietary fiber and reducing sugar content of four kinds of PBMs are shown in Figure 3a,b. In PBMs, neither have a strong relationship with protein composition, in theory. Dietary fiber in soybean products is derived from cellulose, hemicellulose, and lignin in soybean [30], while reducing sugar is derived from the destruction of polysaccharides, including dietary fiber, during extrusion [48].

The findings for three antinutritional factors in PBMs are shown in Figure 3c–e. As these are substances that have negative effects on processing and nutritional characteristics, a lower content of these antinutritional factors in PBMs may improve the quality. The results show that there was no significant relationship between the antinutritional factor levels and the protein composition, with the exception of trypsin inhibitors. HH 43 showed the highest levels of phytic acid (135.62 mg/g) and trypsin inhibitors (16.09 μg/mg), and SF 5 showed the highest phytohemagglutinin content (822.80 pg/mg). Overall, the levels of antinutritional factors for JY 86 and SN 52 were slightly lower.

Trypsin inhibitors mainly exist in 2S globulins [49]. However, excessive intake leads to the decline of protein digestion, absorption, and utilization in the intestine [50]. Phytohemagglutinin mainly exists in 7S [51] and may cause a decrease in digestive ability [52]. Compared with the two other antinutritional factors, the content of phytic acid depends more on the existence of enzymes necessary for its biological process. The key enzymes in phytic acid biosynthesis, myo inositol-3-phosphate synthase (MIPS) and phosphatidylinositol kinase (IPK), are mainly located in 7S [53]. However, the molecular weight of many enzymes is still uncertain and complex [54], and the effect of the 11S/7S ratio on phytic acid metabolism cannot currently be determined. However, the content of antinutritional factors in soybean products changes significantly with the processing process [55]. Therefore, the extrusion process may greatly change the content of antinutritional factors in PBMs.

The composition of isoflavones and the levels found in the four kinds of PBMs are shown in Table 5. Similar to other nutrients, there was no significant correlation between isoflavone content and protein composition. HH 43 showed the highest total isoflavone content (2.005 ng/g), and SF 5 showed the lowest (1.721 ng/g). During extrusion, the bioactivity and stability of isoflavones in PBMs are affected by the processing conditions [56,57]. Isoflavones may degrade, especially at high temperatures or high pressure [58]. One study showed that isoflavones transform into daidzein groups under high temperatures [59]. However, in this study, the daidzein, glycitein, and genistein levels were significantly lower than that of daidzin, glycitin, and genistin, which means that the isoflavones were significantly transformed during extrusion.

The amino acid compositions and contents for four kinds of PBMs are shown in the Appendix A and in Figure 4. Generally, the amino acid compositions of the four kinds of PBMs displayed little difference, but SF 5 was dominant with a higher content of essential amino acids (39.66%). Among all amino acids, the glutamate content was the highest, and it was higher in SN 52 than in the other three (20.08%). As an amino acid that can enhance flavor [60], the high glutamate content may change the flavor of PBMs; analysis of flavor characteristics is required. In addition, we found that the levels of many amino acids increased with a higher amount of 11S, such as threonine, alanine, and arginine; however, some decreased, such as aspartate. This may be due to the differences in amino acid composition between 11S and 7S [61].

Briefly, soybean varieties with a low 11S/7S ratio should not be selected if nutritional characteristics are used as the standard to judge the quality of PBMs.

### 3.4. Effect of Protein Composition on the Flavor of PBMs

The results pertaining to the flavor compounds for the four kinds of PBMs are shown in the Appendix A. Additionally, we plotted the results in Figure 5 according to the classification of substances. As shown, the relative amounts of flavor components in different kinds of PBMs were different, but the main components in all cases were aldehydes, alcohols, alkanes, etc. SN 52 had the highest aldehyde content, and HH 43 had the lowest; the highest alcohol compound content was found in SF 5, and the lowest was SN 52; SN 52 had the highest alkane content, and HH 43 had the lowest. Some of the flavor substances, including acetic acid, hexanal, benzaldehyde, 1-octene-3-alcohol, 2,4-decadienal, trans-2-heptenenal, etc. are defined as typically negative flavor components of soybean products [62], and acetic acid and hexanal are considered to be the main contributors to the beany flavor [63]. In this study, SN 52 showed the lowest beany flavor (10.52%), which means that it should be more accepted by consumers.

There is a relationship between flavor and the protein composition of soybean products, but it is very complex, and even the preparation process is closely related to the flavor [64]. Similarly, the flavor characteristics of PBMs are closely related not only to the composition and structure of the proteins but also to the processing and the characteristics of the products [39]. Compared to protein composition, we believe that the structure of PBMs has a greater impact on flavor. In this study, the structural characteristics of SN 52 demonstrated average features (all indexes were in the middle reaches), which may mean that the soybean varieties suitable for preparing PBMs are not those varieties with a notable difference in their 11S/7S ratios (too high or low). In addition, whether the bad flavor of PBMs can be reduced by adjusting the processing parameters is an important research direction.

## 4. Conclusions

In this study, defatted soybean powder was prepared from four of the main soybean varieties in China, and four kinds of PBMs with different protein compositions were prepared. According to our analysis of the structure, processing characteristics, nutritional characteristics, and flavor characteristics, it may be advisable to prepare PBMs using a soybean with a moderate 11S/7S ratio (1.5:1 to 2.0:1) in order to achieve better quality characteristics. In addition, although the protein composition affects the quality characteristics of PBMs significantly, alterations in the processing process may cause even greater changes, which may play a key role in actual production. This study provides a basic theory for selecting the raw materials for PBMs, and in-depth research would be conducive to further development, including the manufacturing of special equipment, special-use soybean varieties for PBMs, bioactive substances—PBM complex reconstruction, and products with different processing applications. However, these potential applications require a more in-depth study of the production mechanisms of PBMs, including not only the selection of raw materials but also the transformation of bioactive substances, the conformational transition of proteins at different stages, the effects of different additives on the characteristics of PBMs, and even methods of sensory gaining. The above problems need further research to promote the improved development of PBM products.

## Figures and Tables

**Figure 1 foods-11-01112-f001:**
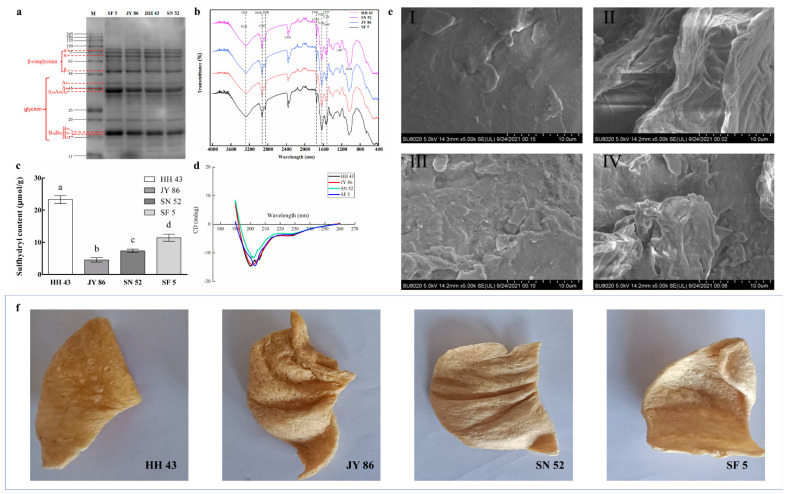
Structural characteristics of four kinds of plant-based meat analogues (PBMs): (**a**) SDS-PAGE of four kinds of PBMs; (**b**) FTIR spectra of four kinds of PBMs; (**c**) Free sulfhydryl content of four kinds of PBMs; (**d**) CD spectra of four kinds of PBMs; (**e**) The microstructure of four kinds of PBMs [I: HH 43; II: JY86; III: SN 52; IV: SF 5]; (**f**) The outward appearances of four kinds of PBMs. Different lowercase letters indicate a significant difference [*p* < 0.05]). HH 43: Heihe 43, JY 86: Jiyu 86, SN 52: Suinong 52, SF 5: Shengfeng 5.

**Figure 2 foods-11-01112-f002:**
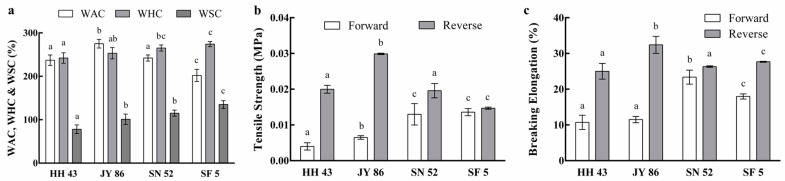
Processing characteristics of four kinds of plant-based meat analogues (PBMs): (**a**) WAC, WHC, and WSC of four kinds of PBMs; (**b**) TS of four kinds of PBMs; (**c**) BE of four kinds of PBMs. Different lowercase letters indicate a significant difference [*p* < 0.05]). HH 43: Heihe 43, JY 86: Jiyu 86, SN 52: Suinong 52, SF 5: Shengfeng 5, WAC: Water-Absorption Capacity, WHC: Water-Holding Capacity, WSC: Water-Swelling Capacity, TS: Tensile Strength, BE: Breaking Elongation.

**Figure 3 foods-11-01112-f003:**
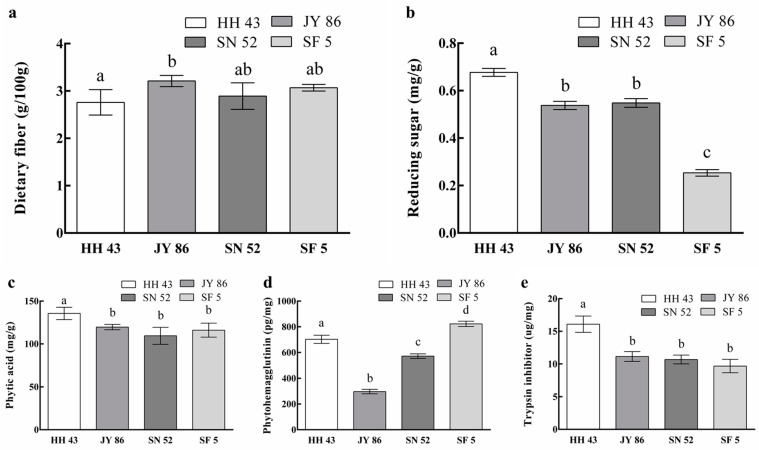
Nutritional properties of four kinds of plant-based meat analogues (PBMs): (**a**) The dietary fiber content; (**b**) The reducing sugar content; (**c**) The phytic acid content; (**d**) The phytohemagglutinin content; (**e**) The trypsin inhibitor content. Different lowercase letters indicate a significant difference [*p* < 0.05]).

**Figure 4 foods-11-01112-f004:**
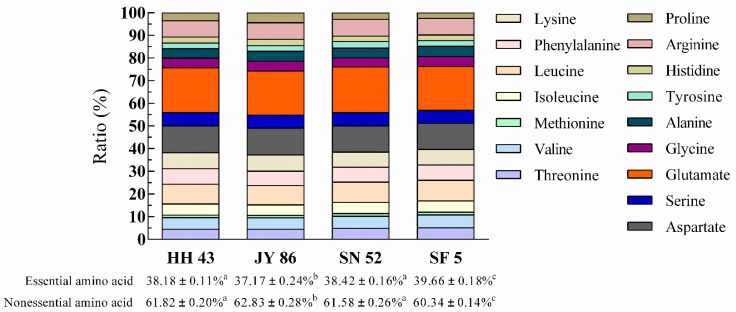
Composition of amino acids (%). Different lowercase letters indicate a significant difference (*p* < 0.05).

**Figure 5 foods-11-01112-f005:**
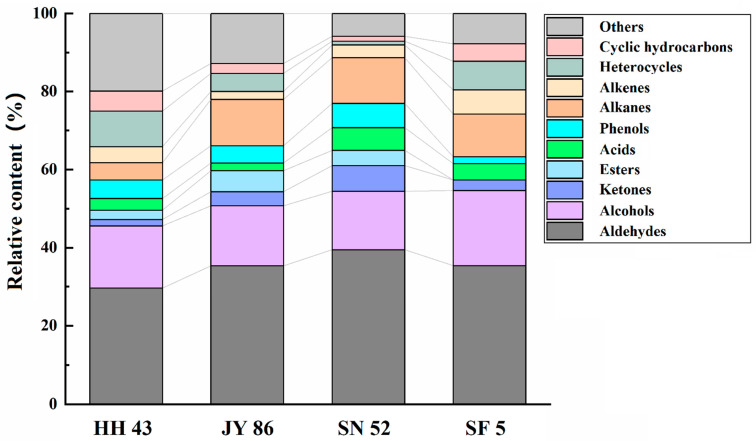
Flavor components of four kinds of plant-based meat analogues (PBMs).

**Table 1 foods-11-01112-t001:** The basic components of four kinds of plant-based meat analogues (PBMs) (%).

	Heihe 43	Jiyu 86	Suinong 52	Shengfeng 5
Protein	46.76 ± 0.07 ^c^	46.68 ± 0.07 ^c^	48.10 ± 0.03 ^b^	49.18 ± 0.04 ^a^
Oil	4.02 ± 0.01 ^b^	3.94 ± 0.04 ^c^	3.97 ± 0.01 ^c^	4.17 ± 0.01 ^a^
Ash	5.28 ± 0.01 ^b^	4.92 ± 0.07 ^d^	5.19 ± 0.05 ^c^	5.46 ± 0.01 ^a^
Moisture	7.26 ± 0.01 ^b^	6.91 ± 0.06 ^c^	7.77 ± 0.01 ^a^	6.24 ± 0.01 ^d^

Different lowercase letters indicate a significant difference (*p* < 0.05).

**Table 2 foods-11-01112-t002:** Subunit composition of the proteins in plant-based meat analogues (PBMs) (%).

Soybeans	α′	α	β	7S	A_3_	A_1a_A_1b_A_2_A_4_	B_1a_B_1b_B_2_B_3_B_4_	11S	11S/7S
Heihe 43	13.53 ± 1.25 ^b^	14.26 ± 0.63 ^b^	7.98 ± 0.05 ^c^	35.77 ± 1.66 ^b^	3.03 ± 0.33 ^d^	17.39 ± 1.24 ^c^	17.37 ± 0.82 ^b^	37.79 ± 0.43 ^d^	1.05 ± 0.05 ^d^
Jiyu 86	11.88 ± 0.85 ^a^	12.02 ± 0.85 ^a^	9.83 ± 0.47 ^a^	33.73 ± 1.32 ^a^	4.69 ± 0.27 ^c^	27.48 ± 0.67 ^b^	20.70 ± 1.73 ^b^	52.87 ± 2.24 ^b^	1.56 ± 0.20 ^c^
Suinong 52	9.69 ± 0.85 ^b^	8.90 ± 0.57 ^c^	8.08 ± 0.35 ^c^	26.67 ± 0.46 ^c^	4.66 ± 0.18 ^b^	24.65 ± 0.55 ^c^	22.19 ± 0.09 ^b^	51.50 ± 0.61 ^c^	1.94 ± 0.05 ^b^
Shengfeng 5	9.62 ± 0.85 ^c^	8.80 ± 0.85 ^b^	9.88 ± 0.14 ^b^	28.30 ± 0.14 ^b^	5.44 ± 0.30 ^a^	37.85 ± 1.53 ^a^	27.30 ± 0.34 ^a^	70.59 ± 2.07 ^a^	2.50 ± 0.11 ^a^

Different lowercase letters indicate a significant difference (*p* < 0.05).

**Table 3 foods-11-01112-t003:** Changes in the secondary structure of four kinds of plant-based meat analogues (PBMs) (%).

	α-Helix	β-Sheet	β-Turn	Random Coil
Heihe 43	24.49 ± 0.19 ^a^	45.72 ± 0.31 ^a^	17.50 ± 0.32 ^b^	12.31 ± 0.15 ^a^
Jiyu 86	25.30 ± 0.19 ^c^	44.98 ± 0.34 ^c^	17.29 ± 0.31 ^c^	12.54 ± 0.15 ^b^
Suinong 52	24.79 ± 0.16 ^b^	45.20 ± 0.32 ^b^	17.32 ± 0.31 ^c^	12.66 ± 0.14 ^c^
Shengfeng 5	24.44 ± 0.18 ^a^	45.12 ± 0.34 ^a^	17.36 ± 0.31 ^a^	12.26 ± 0.16 ^c^

Different lowercase letters indicate a significant difference (*p* < 0.05).

**Table 4 foods-11-01112-t004:** Texture characteristics of four kinds of plant-based meat analogues (PBMs).

	Heihe 43	Jiyu 86	Suinong 52	Shengfeng 5
Resilience	0.32 ± 0.03 ^a^	0.35 ± 0.02 ^a^	0.35 ± 0.03 ^a^	0.39 ± 0.04 ^a^
Springiness	0.80 ± 0.02 ^a^	0.83 ± 0.03 ^a^	0.87 ± 0.04 ^a^	0.92 ± 0.09 ^a^
Hardness (g)	606.29 ± 21.74 ^b^	658.03 ± 19.11 ^a^	693.82 ± 18.88 ^c^	748.10 ± 20.38 ^d^
Adhesiveness (g·sec)	0.15 ± 0.02 ^ab^	0.22 ± 0.06 ^a^	0.25 ± 0.01 ^b^	0.29 ± 0.01^c^
Chewiness	510.34 ± 12.66 ^b^	554.46 ± 12.11 ^a^	578.33 ± 13.53 ^c^	634.01 ± 10.91 ^d^

Different lowercase letters indicate a significant difference (*p* < 0.05).

**Table 5 foods-11-01112-t005:** Composition and content of soybean isoflavones (ng/g).

	Heihe 43	Jiyu 86	Suinong 52	Shengfeng 5
Daidzin	0.413 ± 0.03 ^b^	0.456 ± 0.02 ^a^	0.418 ± 0.02 ^b^	0.389 ± 0.03 ^c^
Glycitin	0.113 ± 0.03 ^b^	0.106 ± 0.02 ^b^	0.118 ± 0.03 ^b^	0.236 ± 0.05 ^a^
Genistin	0.911 ± 0.07 ^a^	0.821 ± 0.08 ^a^	0.877 ± 0.08 ^a^	0.668 ± 0.05 ^b^
Daidzein	0.019 ± 0.01 ^a^	0.017 ± 0.01 ^a^	0.024 ± 0.01 ^a^	0.018 ± 0.01 ^a^
Glycitein	0.432 ± 0.06 ^a^	0.302 ± 0.04 ^b^	0.254 ± 0.05 ^c^	0.312 ± 0.03 ^b^
Genistein	0.117 ± 0.01 ^a^	0.095 ± 0.01 ^b^	0.136 ± 0.01 ^a^	0.098 ± 0.00 ^b^
Total	2.005 ± 0.13 ^a^	1.797 ± 0.16 ^b^	1.827 ± 0.15 ^c^	1.721 ± 0.10 ^d^

Different lowercase letters indicate a significant difference (*p* < 0.05).

## Data Availability

Raw data can be provided by the corresponding author on request.

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
