# Peer review of "The Protein Composition Changed the Quality Characteristics of Plant-Based Meat Analogues Produced by a Single-Screw Extruder: Four Main Soybean Varieties in China as Representatives"

_foods, 2022, doi:10.3390/foods11081112_

Round 1

Reviewer 1 Report

Dear Authors,

my comments are included in the PDF file.

Author Response

Response to Reviewers

Reviewer #1:

  1. Change “Plant-based meat” to “Plant-based Meat Analogues”

Response: Thank you for your valuable comments. We had revised the words and proofread them.

  1. Please describe the research background, hypothesis and aim. Abstract presents brief summary of results. Over all finding of the study should be mentioned at end of abstract.

Response: Thank you for your detailed comments. We rewrote the abstract.

To describe the research background, hypothesis and aim, we had rewritten this part to “Plant-based meat analogues (PBM) is recognized by consumers increasingly because of its meat-like quality and the healthy diet. The single-screw extruder, which is an important method to process PBM, the characteristics of products are affected by the composition of raw materials directly, but few research focused on it. To explore the effect of protein composition on quality characteristics of PBM produced by a single-screw extruder, ……” in Line 19-25.

To describe the finding of the study, we had rewritten this part to “This study proved that the protein composition of materials did affect the characteristics of PBM products produced by single-screw extruder. To produce PBM with higher quality, the soybeans with a particularly different 11S/7S ratio should not be selected.” in Line 35-39.

  1. move Line 33-35 to conclusion’s section

Response: Thank you for your detailed comments. We had deleted this part in the abstract and moved it to the conclusion’s section.

  1. Authors should carefully go through manuscript and check for grammatical and language errors

Response: Thank you for your valuable comments. We have made some modifications to the language problem in the manuscript. If the current manuscript is not acceptable yet, we are glad to improve the manuscript for this problem constantly.

  1. It is absolutely necessary to describe the food safety aspect, i.e., about GMOs, but also about the allergenicity of soybeans.

Response: Thank you for your valuable comments. We had added a part of the content to describe the food safety, including the GMO and the allergenicity of soybeans (Line 109-119).

  1. No research hypotheses were formulated. Please add them.

Response: Thank you for your valuable comments. We added the research hypotheses as “In this study, to explore the potential impact of soybean protein composition in materials on PBM characteristics, we selected ……” in Line 120-121.

  1. Please provide the technical parameters of the extruder construction.

Response: Thank you for your detailed comments. The technical parameters of the extruder construction had been described in Line 154-156. Because this equipment is simple relatively, besides the temperature and rotational speed, the parameters are only processing capacity, but it has little correlation with this study. So please the reviewer to give more guidance on what parameters need to be added.

  1. Unit writing.

Response: Thank you for your detailed comments. We are very sorry for the mistake. All units had been recalibrated.

  1. FTIR or FI-IR? Pls unify

Response: Thank you for your detailed comments. FT-IR should be the right abbreviation of Fourier Transform Infrared Spectroscopy, and the mistake had been corrected.

  1. There is prevailing scope of improvement of discussion. Considering quality of experimental study done the discussion need to be more technically sound.

Response: Thank you for your valuable comments. To improve the quality of the manuscript, we added some content to the Results and Discussion, which was also required by other reviewers. If you are not satisfied with the current version, we will be glad to continue to modify it.

  1. Similarly, conclusion is also summary of results, authors should add future prospects, possible applications or challenges in conclusion.

Response: Thank you for your valuable comments. We followed the suggestions of the reviewers and supplemented the conclusion. Meanwhile, we also moved the last part of the abstract to the conclusion according to the suggestion by reviewers.

“This study provided a basic theory for the material selection of PBM, and in-depth research would be conducive to the development, which would include the manufacturing of special equipment, special-use soybean varieties for PBM, bioactive substances - PBM complex reconstruction, and the products with different processing applicability. However, these potential applications require a more in-depth analysis of the production mechanism of PBM, including not only the selection of materials, but also the transformation of bioactive substances, the conformational transition of proteins at different stages, the effects of different additives on the characteristics of PBM, and even the methods of sensory gaining, etc. The above problems need further researches to promote the development of PBM products better.”

Reviewer 2 Report

The manuscript deals with the protein composition changed the quality characteristics of plant-based meat produced by single-screw extruder: four main soybean varieties in China as representatives.

The width of the text must be revised.

The English language must be revised.

Please separate values from units, e.g. “220 ºC” not “220ºC”.

Please number all equations.

Please replace “Mpa” by “MPa”.

Materials and methods

Line 260- “All determinations were conducted three times, and the whole experiment was repeated twice. P<0.05 was considered significant and all results were expressed as mean ± standard deviation (x ± SD), the mean values of these experiments were used in further analyses. All statistical analyses and graphs were performed with Origin Pro 2018 and SPSS (v. 25; IBM, Armonk, NY, USA).”??used test??and post-hoc test??

Color analysis??

Results and discussion

Pictures of each sample??

Line 389- “The texture characteristics of four kinds of PBM were determined by a texture analyzer and the results were shown in Table 4. The results showed that the five indexes were proportional to the content of 11S directly, that is, SF 5 showed the best texture characteristics. Among them, the resilience and the springiness increased gradually, but there was no significant difference (p > 0.05), while the hardness, adhesiveness, and chewiness were significantly different among different varieties (p < 0.05). This showed that 11S globulin plays an important role in the extrusion process. Increasing the content of 11S could improve the quality of texture protein products to a certain extent, which reason should be attributed to the formation of disulfide bonds[34]. The study of Zheng et al. showed the higher content of β sheet and high ratio of 11S/7S would increase the quality of soy protein gel, which also the disulfide bond might be one of the reasons[35].”???effect of the production process (e.g. temperature) and composition of each sample on the texture must be explained in more detail.

Conclusions

Please do not repeat your results and focus on your main conclusions.

Table 4- Please add units.

Figure 3- Please do not start the y-axis title with “The content…” but “Reducing sugar.”.

Author Response

Response to Reviewers

Reviewer #2:

  1. The width of the text must be revised. The English language must be revised. Please separate values from units, e.g. “220 ºC” not “220ºC”. Please number all equations. Please replace “Mpa” by “MPa”.

Response: Thank you for your valuable comments.

The problems of page layout and the width of the text are automatically generated after the manuscript was submitted. Once the article is received, we will revise the manuscript format immediately. Thank you for your reminder.

We have made some modifications to the language problem in the manuscript. If the current manuscript is not acceptable yet, we are glad to improve the manuscript for this problem constantly.

All the equations had been numbered and the units had been modified.

We are very sorry for the mistake. All units had been recalibrated.

  1. Line 260- “All determinations were conducted three times, and the whole experiment was repeated twice. P<0.05 was considered significant and all results were expressed as mean ± standard deviation (x ± SD), the mean values of these experiments were used in further analyses. All statistical analyses and graphs were performed with Origin Pro 2018 and SPSS (v. 25; IBM, Armonk, NY, USA).”??used test??and post-hoc test?? Color analysis??

Response: Thank you for your detailed comments. Because this manuscript is a part of a large study, we forgot to add some parts that are related to this manuscript directly. Other reviewers also raised this question, and we had rewritten this part. Apologize again for our mistakes.

“All determinations were conducted three times at least, and all results were expressed as mean ± standard deviation (x Ì… ± SD). One-way analysis of variance (ANOVA) with Duncan’s test was used to analyze the differences in the properties of four kinds of PBM using IBM SPSS 25.0 (SPSS Inc., Chicago), p < 0.05 was considered significant and all results were expressed as mean ± standard deviation (x Ì… ± SD). All statistical graphs were performed with Origin Pro 2018 (GraphPad Software Inc., San Diego).” (Line 304-312).

  1. Pictures of each sample??

Response: Thank you for your guidance. In the previous version, because there was no obvious difference between the four PBMs by visual inspection, we did not place sample pictures. In the new version, the sample pictures had been supplemented, as shown in Figure 1f. Meanwhile, we added a part of the content to the outward appearances of four kinds of PBM (Line 385-391).

  1. Line 389- “The texture characteristics of four kinds of PBM were determined by a texture analyzer and the results were shown in Table 4. The results showed that the five indexes were proportional to the content of 11S directly, that is, SF 5 showed the best texture characteristics. Among them, the resilience and the springiness increased gradually, but there was no significant difference (p > 0.05), while the hardness, adhesiveness, and chewiness were significantly different among different varieties (p < 0.05). This showed that 11S globulin plays an important role in the extrusion process. Increasing the content of 11S could improve the quality of texture protein products to a certain extent, which reason should be attributed to the formation of disulfide bonds[34]. The study of Zheng et al. showed the higher content of β sheet and high ratio of 11S/7S would increase the quality of soy protein gel, which also the disulfide bond might be one of the reasons[35].”???effect of the production process (e.g. temperature) and composition of each sample on the texture must be explained in more detail.

Response: Thank you for your comments. We wrote a more detailed discussion. From 11S content, temperature and pressure, it explained why SF 5 might have the best texture characteristics (Line 457-485)

“The content of 11S is related to the textural properties of soy products closely. Especially the protein gel, there is a correlation between 11S and textural properties[39]. Increasing the content of 11S could improve the quality of texture protein products to a certain extent, which reason should be attributed to the formation of disulfide bonds[40]. The study of Zheng et al. showed that the higher content of β sheet and high ratio of 11S/7S would increase the quality of soy protein gel, which also the disulfide bond might be one of the reasons[41]. In addition, in the process of soy texturization, the processing conditions would also strengthen the texture characteristics of the product, such as temperature and pressure, etc. Research showed that, in the conventional soy products processing, adjusting the pressure and temperature of material processing would affect the processing characteristics of the products directly[42], which is caused by changes in the solubility, conformation, and protein aggregation of 11S. In the extrusion process, this situation also exists. In the heating conditions, individual subunits within globulins would undergo dissociation, unfolding, and reaggregation to render them more functional, like higher gelation[43]. In essence, the textured protein is another form of gelation, which also need the rearrangement of different protein subunits. The higher content of 11S also leads to the higher strength of the protein aggregates[44], which would be stronger under high pressure[45]. Therefore, the excellent texture characteristics of SF 5 might be due to the high content of 11S, meanwhile, the high temperature and high pressure provided by the extrusion process enhance the rearrangement of protein subunits and the formation of spatial structure.”

  1. Please do not repeat your results and focus on your main conclusions.

Response: Thank you for your meaningful suggestion. Other reviewers raised the same question, and we rewrote a conclusion, including the summary of results, prospects, possible applications, and challenges.

  1. Table 4- Please add units.

Response: Thank you for your valuable comments. We are very sorry for the mistake. The units of Hardness and Adhesiveness had been added. Resilience, Springiness, and Chewiness are both ratios of two parameters, so there is no unit (Line 490-491).

  1. Figure 3- Please do not start the y-axis title with “The content…” but “Reducing sugar.”

Response: Thank you for your detailed comments. This mistake had been corrected.

Reviewer 3 Report

The authors studied the effects of different soybean protein composition (11S/7S ratio from 1:1 to 2.5:1) on the quality characteristics of PBM in this study.

In overall, the sentence structuring in the manuscript can be further improved.

In Section 1, the authors should explain the gel properties of 11S and 7S, this will provide more robust explanation why their hypothesis was to compare the quality characteristics of PBM made from different soybean protein composition.  

Section 2.3 and 2.5.1, the authors should specify/ describe the methodologies from the Chinese national standards, or they could use international standards such as AOAC methods.

Section 2.3.2, please describe how the samples were pre-treated before adding into the Ellman reagent, likewise for Section 2.3.3, 2.3.4 and 2.3.5. For Section 2.3.5, what sort of concentration of fixative solution and drying alcohol were used?

Section 2.4, please write in proper sentences then into a paragraph.  

Section 2.6, please explain what was the extraction temperature when SPME was inserted into the vial? Why do the authors want to warm the PBM at 80C? Any reason?

Section 2.7, I am not agreed with the experimental set up. The whole experiment should be done in triplicate (meaning repeated thrice), and the determinations should conduct twice in each replicate. I would recommend repeating one more replicate.

Line 371 – how do the authors measure the satiety of SF5? What does processing characteristics linked with satiety of the soybean protein?

Section 4, how do the authors gauge for high acceptance? There are no sensory studies being conducted.

Author Response

Response to Reviewers

Reviewer #3:

  1. In overall, the sentence structuring in the manuscript can be further improved.

Response: Thank you for your valuable comments. We have made some modifications to the language problem in the manuscript. If the current manuscript is not acceptable yet, we are glad to improve the manuscript for this problem constantly.

  1. In Section 1, the authors should explain the gel properties of 11S and 7S, this will provide more robust explanation why their hypothesis was to compare the quality characteristics of PBM made from different soybean protein composition.

Response: Thank you for your detailed comments. We had added a part of the content about gel properties of 11S and 7S in Line 92-98. Because according to the suggestions of other reviewers, we had supplemented the discussion on this issue. To avoid repetition, we only briefly explained it in the first part.

  1. Section 2.3 and 2.5.1, the authors should specify/ describe the methodologies from the Chinese national standards, or they could use international standards such as AOAC methods.

Response: Thank you for your guidance. Through our comparison, the standard method of AOAC is consistent with the national standard of China, so we replaced it. We also supplemented the methods which not in AOAC (Line 174-176 & Line 259-270). We will also pay attention to this problem in the process of future experiments. Thank you for your guidance.

  1. Section 2.3.2, please describe how the samples were pre-treated before adding into the Ellman reagent, likewise for Section 2.3.3, 2.3.4 and 2.3.5. For Section 2.3.5, what sort of concentration of fixative solution and drying alcohol were used?

Response: Thank you for your comments. The methods of the above pre-treated samples were added (Line 179-180, Line 190, Line 199-201, Line 204-210) in the new manuscript. The concentration of the fixative solution and drying alcohol been used were also added in Section 2.3.5.

  1. Section 2.4, please write in proper sentences then into a paragraph.

Response: Thank you for your detailed comments. I could not understand the purpose of this comment because there was much content in Section 2.4. I wonder if you mean to simplify some of them. In addition, one of the reviewers asked us to make mortified on this part. So, if you are not satisfied with the new part, please don't mind telling us and we will continue to mortify it. Thank you for your support.

  1. Section 2.6, please explain what was the extraction temperature when SPME was inserted into the vial? Why do the authors want to warm the PBM at 80C? Any reason?

Response: Thank you for your valuable comments. We had added the extraction temperature “80 ℃” at Line 290. There are two reasons for choosing to preheat. First, the fundamental reason for preheating at 80 ℃ is to balance the system, because it needs to be extracted at the same temperature in the later stage. Then, I can understand your question about why this step should be carried out. This is a method we are used to measure PBM. This step is not necessary for the analysis of soybean milk or tofu, but the traditional methods often had poor results when determining the products with less significant flavor just like PBM. So, we contacted Agilent's engineers, and their advice to us was to preheat for a long time and increase the extraction temperature appropriately. So, we used this method. The result is better at this time. We speculated that the reason for this process is that PBM needs to release its volatile flavor components at a higher temperature because we also found that PBM smells very light at RT.

  1. Section 2.7, I am not agreed with the experimental set up. The whole experiment should be done in triplicate (meaning repeated thrice), and the determinations should conduct twice in each replicate. I would recommend repeating one more replicate.

Response: Thank you for your detailed comments. Your suggestion is consistent with that of other reviewers. We did make a mistake in writing this part. Because this manuscript is a part of a large study, we forgot to delete some parts that were unconcerned with this manuscript directly. And we rewrote a new Statistical analysis in Section 2.7 (Line 304-312).

  1. Line 371 – how do the authors measure the satiety of SF5? What does processing characteristics linked with satiety of the soybean protein?

Response: We are sorry that these words had misunderstood the reader. We can choose to delete this sentence (Line 429-430). Thank you for your reminder.

  1. Section 4, how do the authors gauge for high acceptance? There are no sensory studies being conducted.

Response: Thank you for your valuable comments. We wanted to use the word "high acceptance" to represent the lower beany flavor. But through your question, we realized that this may mislead readers. Meanwhile, other reviewers also suggested that we need to revise Section 4. Therefore, we choose to replace this sentence with the following:

“According to the analysis of structure, processing characteristics, nutritional characteristics, and flavor characteristics to prepare the PBM with better quality characteristics, a moderate 11S/7S (1.5: 1 to 2.0: 1) may be more suitable.”

Reviewer 4 Report

Dear authors.

I reviewed your MS entitled "The Protein Composition Changed the Quality Characteristics of Plant-based Meat Produced by Single-screw Extruder: Four Main Soybean Varieties in China as Representatives". The study is well-designed and the results are well-discussed. However, several corrections must be carried out. You can find all the suggestions in the attached file. 

Table 6 is missing and you must support the increment in the aa that you mention. In Fig. 4, the results seem equal. 

Author Response

Response to Reviewers

Reviewer #4:

The comments of the fourth reviewer were all aimed at the non-standard writing, so I chose to accept and modify all the questions raised by him in the manuscript. Thank you very much for your detailed suggestions.

In addition, because the difference in Figure 4 is not very large, we uploaded the results of amino acid composition as a supplementary material (S1). Finally, A supplement on the importance of threonine, alanine, and arginine was required by the reviewer. Although the content of these three amino acids increased with the high 11S/7S ratio, the range was very small. At the same time, there was no relevant research on the effect of the increase of the content of these three amino acids on the processing characteristics. Moreover, two of them are non-essential amino acids, and it seems that their small increase did not have a substantial impact on nutritional characteristics. Therefore, in order to avoid ambiguity, we want to apply not to supplement this part for the time being.

Round 2

Reviewer 1 Report

Authors answered my question correctly, so I believe that this MS can be published in the present form. 

Author Response

Dear Reviewer

Thank you for all your valuable comments and continuous support.

Reviewer 2 Report

Figure 2a- y-axis, please replace “WCS” by “WSC”.

Figure 2b- y-axis, please replace “Mpa” by “MPa”.

Table 4 caption- Please replace “Table 4. The Texture characteristics of four kinds of PBM” by “Table 4. Texture characteristics of four kinds of PBM”.

Author Response

Dear Reviewer

Thank you for all your valuable comments and continuous support.

We are very sorry for ignoring the errors in the figures and tables. Now we had revised them.

Reviewer 3 Report

Changes have been made to acceptable levels for publication.

Author Response

(The authors gave the same response as above.)
